# Variation in structural motifs within SARS-related coronavirus spike proteins

**Francesca R. Hills**[1]☯, **Alice-Roza Eruera**[1]☯, **James Hodgkinson-Bean**[1], **Fátima Jorge**[2], **Richard Easingwood**[2], **Simon H. J. Brown**[3], **James C. Bouwer**[3], **Yi-Ping Li**[4], **Laura N. Burga**[1], **Mihnea Bostina**[1,2]*

**1** Department of Microbiology and Immunology, University of Otago, Dunedin, New Zealand, **2** Otago Microscopy and Nano Imaging Unit, University of Otago, Dunedin, New Zealand, **3** ARC Centre for Cryo-electron Microscopy of Membrane Proteins, University of Wollongong, Wollongong, New South Wales, Australia, **4** Institute of Human Virology and Zhongshan School of Medicine, Sun Yat-sen University, Guangzhou, China

☯ These authors contributed equally to this work.
* mihnea.bostina@otago.ac.nz

**Data Availability Statement:** The maps and models for the cCoV-007, cCoV-SZ3, and bCoV-WIV1 spikes have been deposited in the Electron Microscopy Data Bank (EMDB), https://www.ebi.ac.uk/emdb/ and to https://www.rcsb.org/ with the

## Abstract

SARS-CoV-2 is the third known coronavirus (CoV) that has crossed the animal-human barrier in the last two decades. However, little structural information exists related to the close genetic species within the SARS-related coronaviruses. Here, we present three novel SARS-related CoV spike protein structures solved by single particle cryo-electron microscopy analysis derived from bat (bat SL-CoV WIV1) and civet (cCoV-SZ3, cCoV-007) hosts. We report complex glycan trees that decorate the glycoproteins and density for water molecules which facilitated modeling of the water molecule coordination networks within structurally important regions. We note structural conservation of the fatty acid binding pocket and presence of a linoleic acid molecule which are associated with stabilization of the receptor binding domains in the "down" conformation. Additionally, the N-terminal biliverdin binding pocket is occupied by a density in all the structures. Finally, we analyzed structural differences in a loop of the receptor binding motif between coronaviruses known to infect humans and the animal coronaviruses described in this study, which regulate binding to the human angiotensin converting enzyme 2 receptor. This study offers a structural framework to evaluate the close relatives of SARS-CoV-2, the ability to inform pandemic prevention, and aid in the development of pan-neutralizing treatments.

## Author summary

SARS-related coronaviruses pose a significant threat to human health and prosperity by virtue of their pandemic potential. Transmission of SARS-related coronaviruses from animals to humans risks another pandemic like the recent COVID-19, or the MERS and SARS outbreaks. To better understand the risk these viruses pose to humans, SARS-related coronaviruses from animal origin should be structurally studied to assess key features related to pathogenicity and stability. The spike proteins from two civet coronaviruses (cCoV-007 and cCoV-SZ3) and a horseshoe bat coronavirus (bCoV-WIV1) were

accession codes EMD-41150, EMD-41152, EMD-41149 and 8TC1, 8TC5, 8TC0, respectively.

**Funding:** YL supported by The Science and Technology Plan Project of Guangdong Province, funded previous work which provided rationale for proteins selected in this study and facilitated project advising. MB supported by Maurice Wilkins Centre for Molecular Biodiscovery New Zealand-China Catalyst program, funded the protein production, travel costs, data collection and funded LNB salary. MB supported by Ministry for Business Innovation and Employment Catalyst program, funded the travel costs and data collection. ARE supported by Division of Sciences, University of Otago, Researcher Fellowship, funded the travel and transport of specimens and funded ARE's salary. The funders had no role in study design, data collection and analysis, decision to publish, or preparation of the manuscript.

**Competing interests:** The authors have declared that no competing interests exist.

structurally resolved by cryo-electron microscopy and analysed for similarity to SARS-CoV-2, the causative agent of COVID-19, for which bats and civets have both been implicated as potential hosts. The quality of the reconstructions allowed us to determine important features of the spikes, such as the network of water molecules that coordinate and stabilize the fatty acid in the fatty acid binding pocket, which are typically not visible in glycoprotein reconstructions. We also identified a different conformation of a key loop in the receptor binding domain of these spikes. This loop has previously been identified as being involved in host recognition and receptor binding. This study offers a structural insight into spike protein structural features which promote success in coronaviruses that infect bats and civets, and helps us to better understand what drives the success of these viruses in such a broad host range.

## Introduction

Genetic and epidemiological studies assessing the origin of SARS-CoV-2 point to zoonotic emergence [1–4], consistent with all six previously existing human coronaviruses [5]. Although the SARS-related coronavirus (SARSr-CoV) bat RaTG13 has the greatest genomic similarity to SARS-CoV-2 (97.8%), the exact animal host or hosts is still debated [3,6]. To date, only a limited number of spike proteins from animal sarbecoviruses have been characterized [6–8], which hinders our understanding of the evolution and origin of the SARS-CoV-2 spike protein, particularly as SARS-CoV-2 almost certainly emerged from an animal sarbecovirus-to-human transmission pathway. Further, considering that three coronaviruses emerged zoonotically in the last 20 years (MERS, SARS-CoV, and SARS-CoV-2), structural research into their closest genetic relatives is a matter of pandemic preparedness [5,9].

SARSr-CoV spike proteins cover the coronavirus membrane and are exposed to the solvent environment in preparation for contacting a potential host cell, undergoing a suite of conformational changes to bind the viral receptor known as angiotensin converting enzyme 2 (ACE2, in the case of SARS-related coronaviruses) and mediate membrane fusion. Each spike is composed of three identical monomers containing key domains involved in restructuring from the pre- to post-fusion state as well as in receptor binding [10,11]. These include the S1 receptor binding domain (RBD) and N-terminal domain (NTD), the S2 fusion protein (FP), heptad repeats, and the transmembrane domain (TM), among others. While a number of studies report detailed structural variations in SARS-CoV-2 spike proteins, which is the most immunogenic component of the virus and forms the basis of many of the anti-SARS-CoV-2 mRNA vaccines, less attention has been devoted to the architectures of SARSr-CoV spikes [12]. Such information is essential for the understanding of SARS-CoV-2 emergence, which likely occurred through a series of animal-to-human jumps via an as-yet unknown intermediate animal host [2,13]. As structure informs function, it is also important to elucidate more SARSr-CoV spikes to identify coronaviruses that may be capable of triggering future zoonoses and therefore pose a potential risk.

Previous studies on SARSr-CoV glycoproteins have confirmed the presence of linoleic acid within the fatty acid binding pocket (FABP) of SARS-CoV, SARS-CoV-2, and pangolin CoV-GX [8,14] where it stabilizes the receptor binding domain (RBD) in the "down" position and prevents premature activation and release of the S1 subunit [14,15]. Little detailed information exists on how the fatty acid is stabilized within this pocket, namely the water molecule coordination network that likely exists within the FABP and allows for the highly-ordered position of the acid previously observed [12,13]. Understanding how this acid is stabilized in

the pocket would better contribute to our ability to manipulate this feature of the coronavirus spike by way of ligand or inhibitor targeting. In addition, there seems to be little structural information available on the water network between the spike protein and its numerous glycosylations which decorate the exterior of the protein towards the solvent-exposed area [14,15]. As glycoproteins can obscure antibody binding epitopes and are often highly conserved, knowledge on the post-translational modifications of SARSr-CoVs may offer insight into the immunogenicity of glycoproteins and leave clues to their evolutionary history [16,17].

To elucidate the structures of some SARSr-CoVs with relevance to SARS-CoV-2, we selected three spike proteins from a bat and two civet coronaviruses and subjected them to cryo-EM structural analysis. Here we report three novel structures of SARSr-CoV spike proteins belonging to bat SARSr-CoV WIV1 (hereafter WIV1), and civet SARSr-CoV SZ3 (hereafter SZ3) and SARSr-CoV 007 (hereafter 007), at resolutions of 1.88 Å, 1.92 Å, and 2.10 Å, respectively. As bats are the natural host of SARSr-CoVs [18] and civets are potential intermediate hosts for SARS-CoV-2, this research narrows the structural knowledge gap on SARSr-CoV spikes, and informs future investigations into the origin of SARS-CoV-2.

## Results

### Structural conservation amongst SARr-CoV glycoproteins

Coronavirus spikes must maintain a stable conformation without proceeding to the post-fusion state until receptor recognition has occurred. Several structural motifs were proposed to regulate this behavior in the case of SARS-CoV-2 spike, including the D614 substitution, the fatty acid binding ability of the spike, and the trimer interface [19–22]. We have chosen three SARSr-CoVs from evolutionarily important hosts, namely of civet and bat origin given both these hosts are key suspects as intermediate hosts of SARS-CoV-2, in order to determine how structural variations in these motifs regulate stability of the glycoproteins.

We have solved the structures of 007, SZ3 and WIV1 viral glycoproteins (Figs 1 and S1) and compared them against the spike proteins from SARS-CoV-2 WT (PDB: 6VXX), bCoV RaTG13 (PDB:6ZGF), pCoV GX (PDB: 7CN8), and pCoV GD (PDB: 7BBH) (S2–S6 Figs). The SARSr-CoV glycoproteins follow the same general architecture with a root mean square deviation (RMSD) values between 4.0 Å and 4.4 Å when overlaid with SARS-CoV-2, indicating a high degree of overall structural conservation, although local structural variations exist in the S1 domain. Of the spikes solved in this study, the NTD, RBD and receptor binding motif (RBM) of WIV1 share the most structural similarity with SARS-CoV-2, with RMSD values 1.6 Å, 1.1 Å and 1.7 Å, respectively (S4, S5 and S6 Figs). Data collection and refinement statistics can be found in S1 Table.

Interestingly, the residues 470–491 in the RBM, a region we've described as the 'scoop loop', adopts a structurally distinct conformation in these spikes when compared with SARS-CoV-2 variants or the other reported SARSr-CoVs spikes (Fig 1B and 1C) [23–25]. This loop is known to be involved in ACE2 receptor binding in Omicron variant BA.1 and RaTG13 [24,26]. Geneious alignment of the sequences from key bat, pangolin, civet and human coronaviruses revealed that several scoop loop residues are consistently conserved across host species, particularly an X/P/C sequence where X is either a K or a T, and a N/C/Y sequence which is conserved across all species analyzed (Fig 1I). Several non-conserved residues within an 8-residue N-terminal stretch have altered the hydrogen bonding network that coordinates the tertiary structure of the scoop loop (Fig 1B, 1C and 1I), resulting in a shift of up to 3.4 Å, compared to the same loop in RaTG13 and SARS-CoV-2, which share a conserved fold.

In addition, structural variation is observed within the fusion protein domain between the current SARSr-CoV spikes and SARS-CoV-2 with respect to the loop containing residues

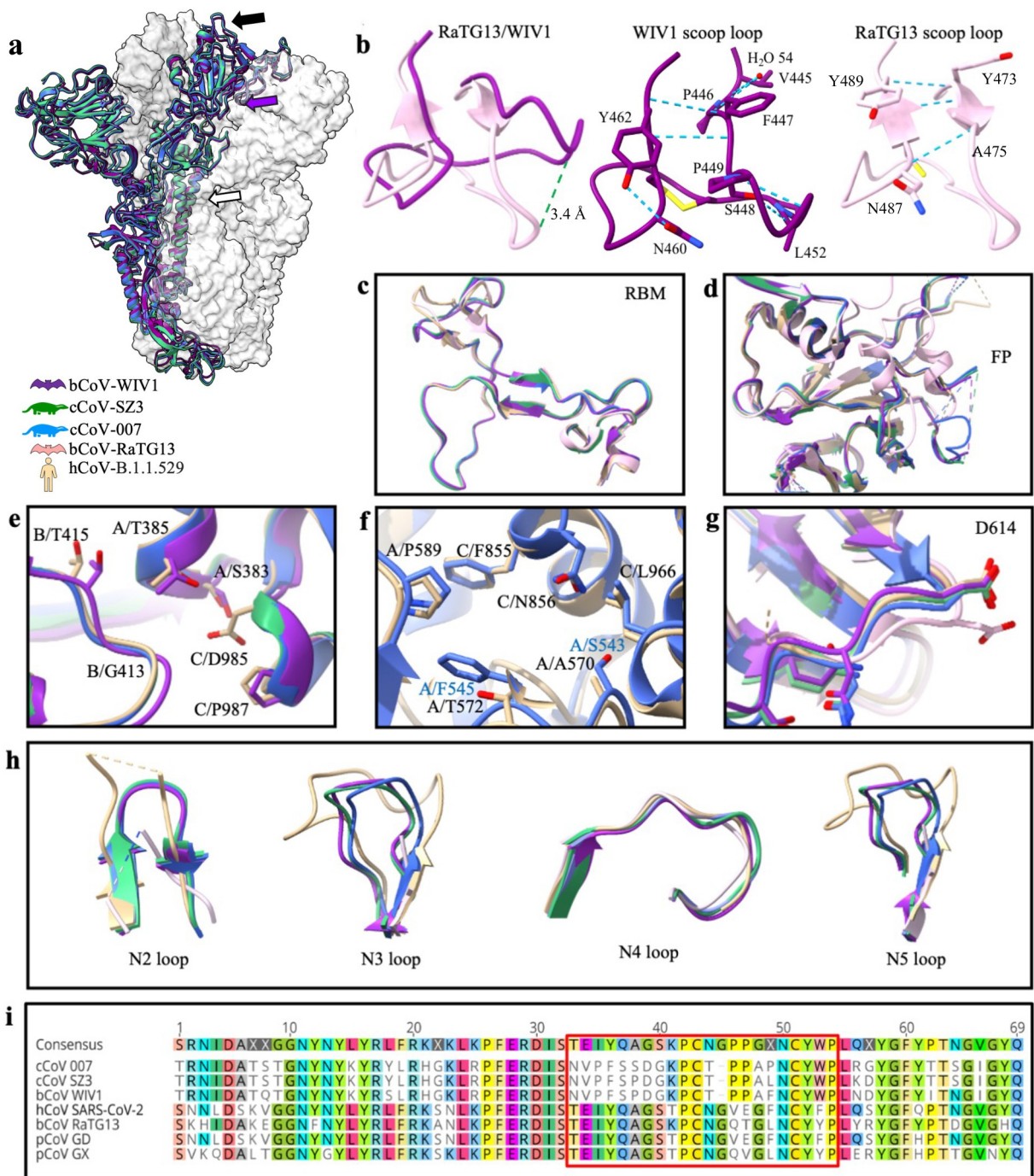

**Fig 1. Structural and sequence-based comparison of S1 domains between SARSr-CoV spike proteins.** *(a)* Structural alignment of the spike, cCoV-SZ3 (green), cCoV-007 (blue) and bCoV-WIV1 (purple), other monomers displayed in gray as a surface representation. Black arrow indicates receptor binding motif (RBM), purple and white arrows indicate important trimer interfaces. *(b)* Protein alignment of RBM loop of interest between RaTG13 (pink) (PDB: 6ZGF) and bCoV-WIV1. RMSD calculated to 3.4 Å. Bond lengths measured in ChimeraX and hydrogen bonds indicated by dashed blue line. *(c)* Protein alignment of the glycoproteins against RaTG13 and SARS-CoV-2 WT (tan) (PDB: 7QUS) at RBM and *(d)* fusion peptide (FP). *(e-f)*. Key residues at the trimer interface involved in structure guided stabilization of SARS-CoV-2 in closed conformation compared to the glycoproteins. *(g)* Protein alignment of the glycoproteins against RaTG13 and SARS-CoV-2 WT with D614 side chain displayed. *(h)* Protein alignment of four N-terminal domain (NTD) loops in the glycoproteins against RaTG13 and SARS-CoV-2 WT. *(i)* Geneious global amino acid sequence alignment of RBM from 6 glycoproteins, bCoV-WIV1, cCoV-SZ3, cCoV-007, human (h) SARS-CoV-2 WT, bCoV RaTG13, pangolin (p) CoV GX (PDB: 7CN8) and pCoV GD (PDB:7BBH). Scoop loop outlined in red.

V620—V642 (Fig 1D). Differences are present in the region adjacent to the D614G mutation site, which has been shown to correlate with increased infectivity and was present in all SARS-CoV-2 variants [27–30]. Comparison of the SARS-CoV-2 D614G site with SARSr-CoV glycoproteins showed a shift of 2.3–2.8 Å in the main chain (Fig 1G). Although the SARSr-CoV glycoproteins have an aspartic acid at position 614, the arrangement of the flanking residues is conserved with SARS-CoV-2 (Fig 1G).

The interface between individual monomers is important in the structural stability of the spike and was investigated in several studies of SARS-CoV-2 variants [31–33]. There is considerable structural conservation between several studies of SARS-CoV-2 variants [28–30]. There is also considerable structural conservation between the solved glycoproteins of animal origin and the SARS-CoV-2 structure (Fig 1E and 1F). Although residues S380 and D982 are not conserved between SARS-CoV-2 and the glycoproteins at one of the trimer interfaces, the distance between interface contacts is comparable across spikes (<1 Å shift), with the exception of residues D982 and S380, which were 5.9–6.1 Å further away from each other in the solved glycoproteins. Another interface includes non-conserved residues A570, T572 and K856, but no more than a maximal 1.9 Å shift is observed.

The NTD loops N1-5 from SZ3, 007 and WIV1 were compared to RaTG13 and the disordered loops of SARS-CoV-2 (Fig 1H). In all these glycoproteins, the N2, N3, N5 loops are shorter than in RaTG13, consistent with research showing large variations in NTD length polymorphisms across the sarbecovirus family, indicating frequent loop remodeling [20,34–38].

## Biliverdin-binding pocket within N-terminal domain

In all the cryo-EM maps, a density was identified within the hydrophobic biliverdin-binding pocket (BBP) formed by the beta-sandwich of the NTD (Fig 2). This suggests the presence of biliverdin or bilirubin, as these metabolite molecules have been reported to bind the BBP in other coronavirus spike structures [39,40]. When biliverdin is modeled into this unassigned density, each tetrapyrrole group nests within the pocket and a putative hydrogen bond could potentially be adopted between a carbonyl group of a lactam ring and N105 of the protein, which is conserved across all three structures. There is no density for the two solvent-exposed

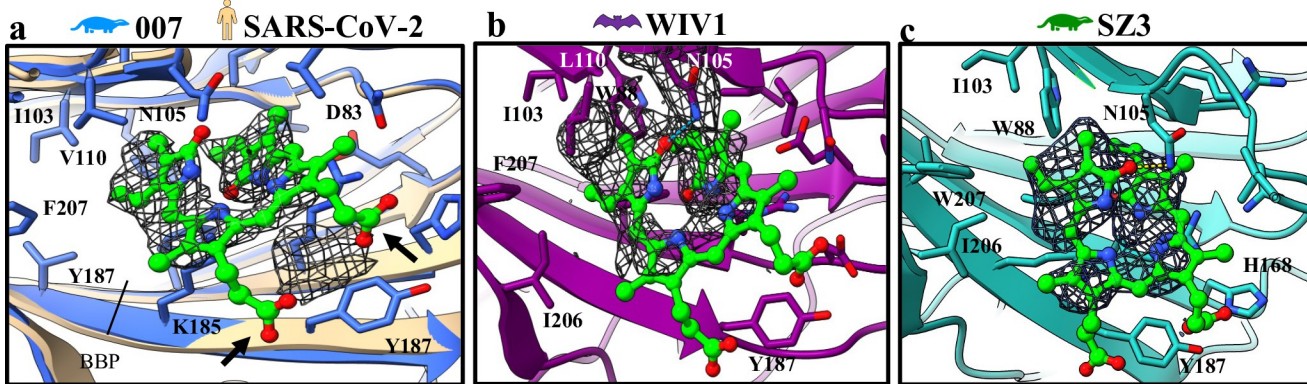

**Fig 2. Biliverdin-binding pocket of SARSr-CoV glycoproteins.** The hydrophobic biliverdin binding pocket is formed by the NTD beta sandwich. The corresponding density is rendered as a mesh within the pocket. Residues within 5 Å of biliverdin are displayed. (*a*) The 007 structure is overlayed with a 1.8 Å reference structure (7B62, tan) known to be bound to biliverdin (lime, molecule from 7B62). Black arrows indicate solvent-exposed carboxyl groups which lack density in the electron maps of all three spikes. (*b*) The biliverdin density in WIV1 is continuous with the density for some of the pocket residues, namely L110 and N105. (*c*) The SZ3 map possesses the most isotropic biliverdin density. All SARSr-CoV spikes share conserved Y187 and N105 residues.

carboxylate groups of the metabolite, potentially due to a lack of protein contacts to stabilize the group, or due to negative Coulomb potential within these residues. We note a lack of density was also seen with some negatively charged amino acid side chains in the NTD observed in these structures (e.g. E161, D153, D195), suggesting decarboxylation of the structure by the electron beam during data collection.

## Water molecules in the fatty acid binding pocket

The fatty acid binding pocket (FABP) within the RBD of SARS-CoV-2 glycoproteins has previously been reported to hold a 9-octadecenoic acid (linoleic acid) [8,14]. Several glycoprotein structures with resolution of 2.5 Å or better have been deposited to the PDB containing a fatty acid modeled in this pocket (7QUR, 7QUS, 7ZH1, 7CN8). Previous studies have also demonstrated that the presence of a fatty acid in the FABP is responsible for regulation of the "up" or "down" conformation paradigm of the RBD, whereby presence of the acid ensures the RBD remains in the 'down' conformation and loss of the acid appears to permit adoption of the 'up' conformation [14,15,41]. These structures reveal the presence of linoleic acid in the FABP of all three SARSr-CoV glycoproteins (Fig 3).

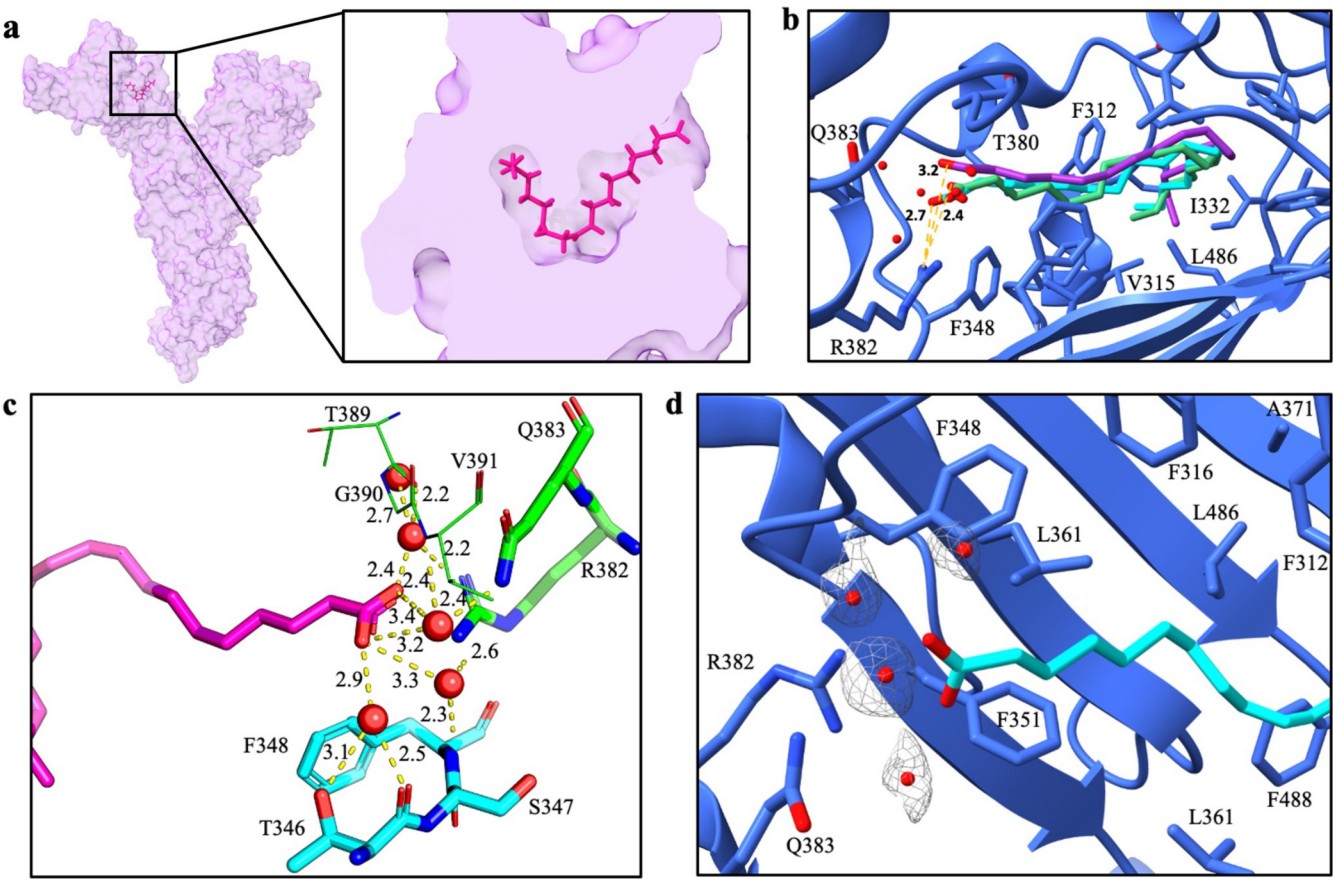

**Fig 3. Fatty acid binding pocket.** *(a)* Relative location of fatty acid (9-octadecenoic acid, magenta) within the FABP of bCoV-WIV1 spike protein. *(b)* Overlay of cCoV-007 (green), cCoV-SZ3 (cyan) and bCoV-WIV1 acids (purple). Pocket residues within 4 Å are displayed in stick representation. *(c)* Water molecule coordination between the linoleic acid and chain A of the cCoV-SZ3 spike is represented by a yellow dotted line. Donor-acceptor hydrogen bond cut-offs were set to 3.4 Å. Waters are displayed as red spheres and bond lengths are reported in angstroms. *(d)* Water molecule densities are shown, within the FABP of cCoV-SZ3.

Remarkably, the maps reveal a conserved network of water molecule coordination of the fatty acid inside the FABP. These waters support the acid within the pocket, in addition to a single putative hydrogen bond formed between the carboxylate head of the linoleic acid and residue R382 of the spike protein (Fig 3B), at distances of 2.4 Å (SZ3), 2.7 Å (007) and 3.2 Å (WIV1).

## N-linked glycosylation of SARSr-CoV spike proteins

All three maps contain well-resolved isotropic densities for complex oligocarbohydrate trees which decorate the exterior of the spike proteins (Fig 4). The densities guided modeling whereby carbohydrates were modeled based on fit. These trees are composed of N-linked glycosylations, of which various arrangements of fucose (Fuc), mannose (Man) and/or beta-D-mannopyranose (Nag) molecules branch. There are 17 glycosites per monomer in WIV1, 16 in SZ3 and 15 in 007 (Fig 4B). This is supported by 19 N-linked glycosylation sequons present in WIV1, 18 in 007 and 17 in SZ3. While 22 N-linked glycosylation sequons are present per monomer in SARS-CoV-2 WT (PDB: 6VXX) with 16 resolved through cryo-EM [42]. Generally, the majority of glycosites are conserved across the SARSr-CoV spikes, but their oligocarbohydrate composition varies significantly.

All reported maps reveal polar interactions between N-linked glycans and nearby water molecules or amino acid side chains. For example, in the SZ3 spike structure, the oxygen atoms of the fucose molecule on the N770 glycosylation (Fig 4G) form putative hydrogen bonds with the side chains of S901 and Q904, stabilizing the carbohydrate tree. Further, the glycosylation on N145 coordinates two water molecules between the fucose and a main chain carbonyl group of the glycoprotein. Within individual glycan trees, each saccharide addition is further stabilized by hydrogen bonding with the preceding sugar. These interactions reveal how the protein and its hydration shell contribute to the conformations adopted by glycan modifications, and may allow glycans to adopt a locally stable conformation which might assist in the ability of glycans to obscure essential antibody-recognition sites. Although there are three N-linked glycosites conserved within the RBD of the SARSr-CoV spikes, they are distant from the receptor binding motif when in the closed conformation, with the closest glycan tree approximately 9 Å from the nearest RBM residue (I415).

## Discussion

In recent years, numerous structural and functional studies regarding SARS-CoV-2 spike protein have been published [42–45]. However, it remains challenging to evaluate its origin, the risk of future SARS-CoV outbreaks, or to support the design of broadly neutralizing coronavirus antibodies. Here we have expanded the gallery of SARSr-CoV spikes in order to better understand the variability of structural motifs involved in receptor binding, spike stability and antibody recognition. SARS-CoV was demonstrated to have originated in fruit bats and transmitted to humans via an intermediate civet host, under circumstances similar to that of SARS-CoV-2. SARS-CoV-2 shows high homology to other viruses of bat origin (e.g. RATG13) and likely also transmitted via an intermediate animal host, but that host currently remains unknown. Civets have remained a likely suspect, among other potential animals like racoon dogs and bamboo rats [13]. Alignment analysis revealed other CoVs of bat origin were more divergent from SARS-CoV-2, but SZ3 is genetically close to SARS-CoV (GD03T0013) and the first epidemic SARS coronavirus GZ02 [46]. Coronavirus SZ3, which originated in bats and transmitted into civets sometime in the 1990s, is thought to be the result of a recombination event of bat CoV strains Rp3 and Rf1, with the recombination breakpoint at the nsp16/spike [47]. Both SZ3 and WIV1 were neutralized by several monoclonal antibodies [48]. WIV1 has

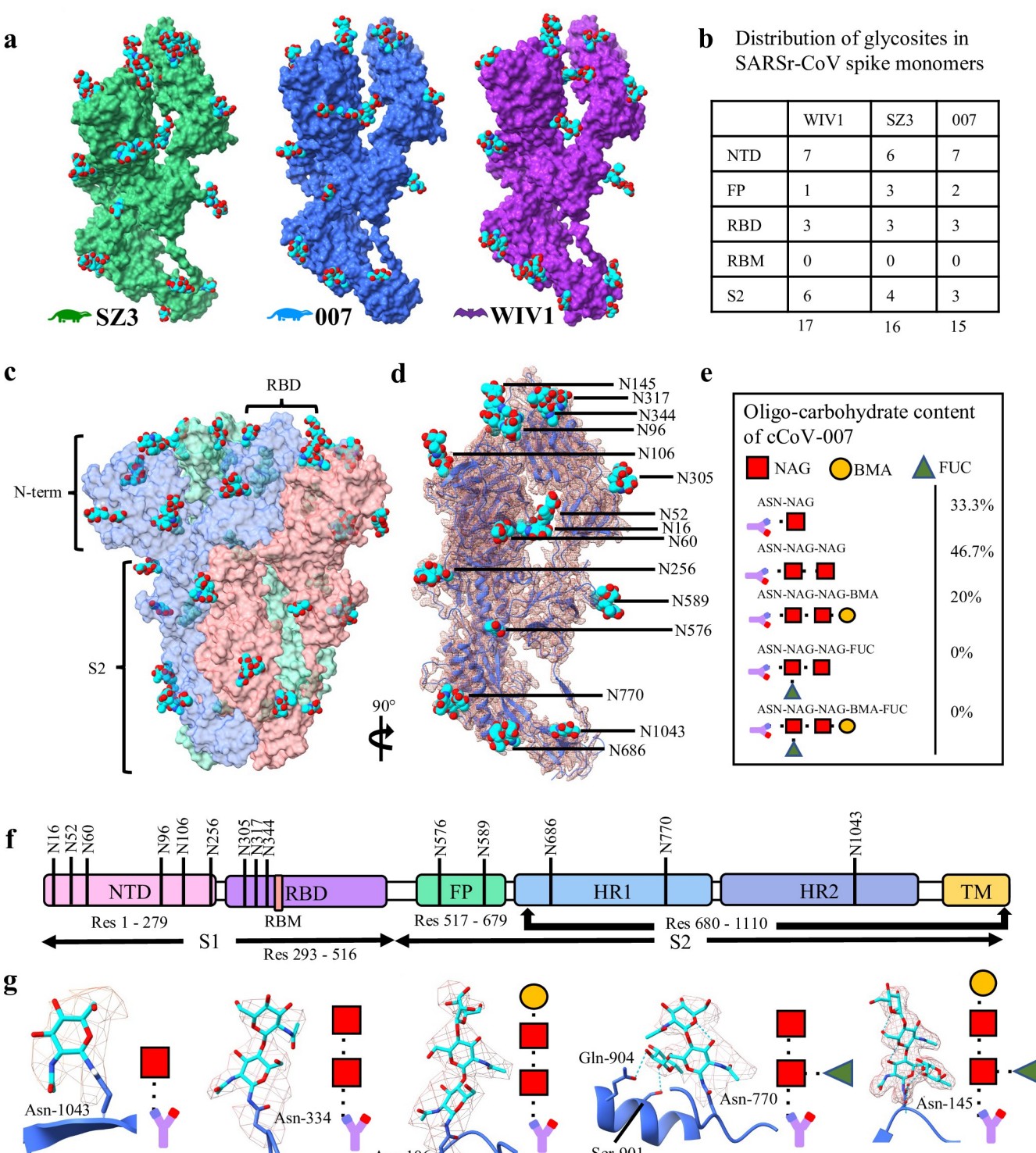

**Fig 4. Extensive N-glycosylation of SARSr-CoV glycoproteins.** *(a)* Monomer of cCoV-SZ3, cCoV-007 and bCoV-WIV1 are shown with glycans displayed in cyan. *(b)* The distribution of glycosites varies between the three SARSr-CoV spikes across the main architectural domains, except within the RBD. *(c)* Trimeric cCoV-007 spike is represented with molecular surface display and coloured by chain. All 45 N-glycans are displayed as cyan spheres regardless of carbohydrate composition. *(d)* Chain A is displayed nested within the Coulomb density map. *(e)* Carbohydrate tree compositions (based on density-directed modelling) are reported for cCoV-007. *(f)* Each glycosylation site is mapped to the domain organization of the cCoV-007 glycoprotein. Although there are two N-linked glycosylations within the receptor binding domain, they are both distant from the receptor binding motif, with the closest glycan approximately 9 Å from the nearest RBM residue (Ile-415). Schematic not to scale. *(g)* A representative carbohydrate from each type of complex N-glycosylation shown with the Coulomb density map displayed as a mesh. The N-linked asparagine and other relevant neighbouring residues are labelled.

been proposed to pose a significant threat via direct transmission to humans, as both full-length and chimeric WIV1-CoV replicated efficiently both in human airway cultures as well as *in vivo* in BALB/c mice models [49]. Further, 007 and WIV1 are recognized by antibodies isolated from SARS-CoV-2-infected patients [50,51]. These factors influenced our decision to select these three strains for structural study.

Previous research suggests that presence of linoleic acid in the FABP regulates the conformation of the RBD, stabilizing the "down" conformation of the spike in which binding to ACE2 does not occur [14,41]. This is supported by multiple *in vivo* studies showing that treatment with linoleic acid reduced the viral load of SARS-CoV-2 on human cells [14,15,41]. The present study confirms the structural conservation of the FABP between human and non-human betacoronaviruses. The fact that all trimers were captured in the RBD 'down' position further supports the role of linoleic acid in stabilizing this conformation and recommends targeted delivery of linoleic acid as a viable treatment avenue against future zoonotically-derived CoV outbreaks. Further, transition of the RBD from 'down' to 'up' has been suggested to be pH-regulated, likely due to a change in protonation state of the side chain nitrogens on a single arginine residue which hydrogen bonds with the acid [52], complementary with other studies that suggest an aspartic acid residue might play a role [53,54]. This is complemented in the present study whereby cryo-EM structures were obtained at pH 8.0 and all RBD's are in the "down" conformation. The high resolution obtained for all the three maps allowed us to visualize the water molecule network within the FABP that coordinates the linoleic acid and their role in the stabilization and subsequent destabilization of the acid. Disruption of the water molecule network, through pH-dependent deprotonation of water-interacting side chains, may result in a weakening of the acid-pocket interaction.

SARSr and SARS-CoV-2 spikes share a high degree of structural conservation, particularly in the S2 and N-terminal domains, suggesting these SARSr-CoV spikes have the potential to be neutralized by several class II anti-SARS-CoV-2 antibodies [55]. This has implications for rapid healthcare responses should 007, SZ3 or WIV1 spill over into humans, although it is possible that class I antibodies, which bind the RBM, may be less efficient at cross-neutralizing SARSr-CoV spikes by virtue of local variation in the Thr 470-Pro 491 loop i.e. Tixagevimab + Cilgavimab [56]. Glycosylations act as shields against the host immune response, as mutations in glycosites can alter antibody susceptibility by sterically hindering access to antibody binding sites. Structure-guided glycosome studies can provide insights into antibody evasion or evolutionary history. For example, NTD glycosite N96 and RBD glycosite N344 are conserved in all three SARSr-CoVs but are not observed in SARS-CoV-2, suggesting that the civet and bat SARSr-CoVs may be more resistant to antibodies that recognize this region. Further, N709, N1134, N1074 and N1098 are glycosites on the distal end of S2 near the transmembrane domain of the SARS-CoV-2 spike and are conserved in WIV1 (except for N1074) and SZ3 (except for N1134). However, 007 lacks all of these sites but N1074. Of the civet spikes, SZ3 may be more closely related to SARS-CoV-2 than 007 based on glycosome comparisons.

Previous work has shown WIV1 to bind ACE2 orthologs and sustain low level infection in human cells [49,57]. During ACE2 binding, residues near the scoop loop contact the N-terminal α1 helix of the hACE2 receptor, playing a role in host specificity and cell entry [58,59]. The WIV1 structure, captured in a near-native state, shows an altered tertiary structure compared to SARS-CoV-2 by virtue of a different hydrogen bond network. Interestingly, the conformation of the scoop loop in the SARSr-CoV structures is different from that of an X-ray crystallography RBD structure of the WIV1 spike (PDB: 7TTY), which indicates there are at least three dynamic conformations this loop can adopt, whilst not precluding the WIV1 spike from binding ACE2 orthologs [49,57]. Both WIV1 and RaTG13 originate from horseshoe bats, suggesting that the bat ACE2 receptor is lenient with respect to the fold of this loop, as at least two

different conformations are permissive for bat ACE2 binding and cell entry, and three are permissive for human ACE2 [23,24,49,57,60,61]. In the absence of a civet ACE2-bound spike structure, it is unknown if civet spikes are evolutionarily restricted by their corresponding civet ACE2 receptors from adopting a similar loop confirmation as that of SARS-CoV-2.

Circulating strains of SARS-CoV-2 contain the D614G mutation, which enhances infectivity and protein stability [27–30,62], favoring the RBD "up" conformation while maintaining stability by preventing premature S2 dissociation due to the removal of a hydrogen bond and salt bridge associated with D614 [58]. The glycoproteins show conservation in the flanking residues near the D614G mutation site but retain the original D614 residue. Like human coronaviruses, we hypothesize that zoonotic spike proteins stand to receive a fitness advantage by obtaining a point mutation at residue 614.

Trimer interfaces (Fig 1E and 1F) have been identified as important players in controlling the conformation of the SARS-CoV-2 spikes. Previous work showed a series of six cysteine mutations resulted in disulfide bond formation that locked the RBD in the down conformation, whilst other four mutations showed increased accessibility to the RBD by encouraging the RBD "up" conformation [31,32]. Triggering this "up" conformation is important in vaccine development, as it allows antibody responses against the ACE2 binding site. Structural variations in this region across the SARSr-CoV-2 seems to play an important role in infectivity, and more research is clearly needed in this area.

Variations in the NTD loops are common within the sarbecovirus family. During SARS-CoV-2 evolution, successive truncations within these loops are associated with increased spike stability and viral entry [33–35,37,38,63]. These loops are also hypervariable in SARS-CoV-2 to allow for immune escape from NTD targeting antibodies [34,37]. Due to the loop's flexibility and hypervariability, structural characterisation has occurred only when epitopes within or near the loops are antibody bound [35]. As the NTD loops are important parts of antibody epitopes, viral entry and stability, the characterization of these loops in zoonotic coronaviruses is important for future antibody treatment and vaccine models.

In all the spikes, the NTD biliverdin binding pocket (BBP) likely harbors either biliverdin or bilirubin, which are present as tetrapyrrole products of heme metabolism due to cellular respiration. Supporting this a previous study showed the presence of biliverdin or bilirubin within this binding pocket [39]. Levels of biliverdin are elevated during coronavirus infection as a result of tissue damage, and are correlated with COVID-19 symptoms and mortality [64–67]. Previous research highlighted that biliverdin or bilirubin can effectively block class II antibodies from binding the NTD, by virtue of a 'gate' on a flexible loop on the distal end of the NTD [39]. Although the amino acid composition of the BBP varies, this gate loop is well conserved between these spikes and that of SARS-CoV-2. We note that binding of these metabolites to the BBP in spikes solved in this study is unlikely to be at high affinity, as our specimen preparation lacked any noticeable green hue as reported by others [39]. Additionally, these spikes share a conserved Y187 residue, which has been previously proposed to weaken binding [39]. Some published cryo-EM structures of SARS-CoV-2 spikes contain equivalent unidentified entities in the BBP (7DF3, 7K43, 6ZP2, 6ZXN, and others) which, like ours, lack density for the carboxyl groups of biliverdin but possess partial density for the lactam rings [68–71]. We suggest that tetrapyrrole-mediated immune escape, as a fitness strategy, extends beyond SARS-CoV-2 and is employed by animal SARSr-CoVs as well.

Despite the emergence of multiple zoonotic coronaviruses which cause severe disease in humans, little structural research has been undertaken on the spike proteins from the SARS-related coronavirus species. This study addresses a gap in knowledge allowing recognition of conserved motifs, as well as providing the structural basis for risk assessment of future cross-species transmission, and the development of pan-neutralizing treatments.

## Materials and methods

### Expression vector constructs

Plasmid design followed previously described methods [70] for the production of trimeric pre-fusion spike ectodomains and were codon-optimized for mammalian cells. Plasmids contained the gene encoding the civet SARS CoV SZ3/2003 (residues 14–1194, GenBank: AY304486.1), civet SARS CoV 007/2004 (residues 14–1194, GenBank: AAU04646.1) and Bat SARS-like coronavirus WIV1 (residues 14–1194, GenBank: KC881007.1) spike protein ectodomains. Gene sequences begin after the hydrophobic N-terminal signal peptide sequence (SDLD or YTIE) and end before the transmembrane domain (QYIK). An N-terminal 5' flanking extension was added containing a BamHI restriction site (GGATCC), Kozak consensus sequence (GCCACC), and μ-phosphatase derived signal peptide (MGILPSPGMPALLSLVSLLSVLLMG CVAETGT) to ensure protein extracellular secretion. Additionally, a C-terminal 3' flanking extension was added containing a TEV cleavage site (GSGRENLYFQG), a T4 foldon trimerization motif (GGGSGYIPEAPRDGQAYVRKDGEWVLLSTFL), Avi-tag (GSGGLNDIFEAQ-KIEWHE), 8×His-tag (GSGHHHHHHHH), stop codon (TAA), and EcoRI restriction site (GAATTC). The pcDNA3.1+ plasmids containing the recombinant spike proteins with 3' and 5' flanking extensions were obtained through Genscript Hong Kong.

### Bacterial transformation

Plasmid DNA concentration was amplified in competent cell line *E. coli* DH5α using standard bacterial transformation methods [72] using QIAPrep miniprep and Qiagen midiprep kits as per manufacturer's instructions [73], eluting into TE buffer for long-term storage.

### Plasmid transfection

Plasmids containing genes for the spike proteins were transfected in an ExpiCHO Transfection System (Thermo-Fisher Scientific). ExpiCHO-S cells were maintained at 37˚C in 8% $CO_2$ at ≥80% relative humidity. Transfections were carried out as per the manufacturer user guide (Cat No: A29133, Publication Number: MAN0014337). The supernatant which contained recombinant glycoproteins was harvested at day 4 and day 8, and pooled together. Glycoprotein expression was confirmed by SDS-PAGE gel and western blot, probing for the His-tag with anti-6×His-Tag monoclonal antibody (Thermo-Fisher Scientific, MA1-21315) and using goat anti-mouse IgG (H+L) cross-adsorbed secondary antibody dylight 800 (Thermo-Fisher Scientific, SA5-10176). Western blots were imaged on an Odyssey Fc imaging system.

## Protein purification and concentration

His-tagged SARSr-CoV spike proteins were purified by immobilized affinity chromatography using Ni-NTA agarose beads (Qiagen) using a gravity flow column (Thermo-Fischer Scientific). Protein was eluted in a buffer containing 50 mM Tris-HCl pH 8.0, 200 mM NaCl, 300 mM imidazole and dialyzed through an 50,000 molecular weight cut-off (MWCO) ultra centrifugal device (Amicon) to remove the imidazole. Sample purity was assessed by SDS-PAGE and western blot as described above. Prior to electron microscopy (EM) studies, samples were dialyzed into a buffer containing 20 mM Tris-HCl pH 8.0, 100 mM NaCl and concentrated to 0.5 mg/mL using an Amicon centrifugal concentrator with 50,000 MWCO.

### Negative-stain EM

Specimens were assessed for sample quality and purity by negative-stain EM on a Philips CM-100 transmission electron microscope (TEM). Copper 300 mesh carbon coated grids were

negatively plasma discharged in a GloQube discharge system for 30 seconds at 15 mA prior to the application of 4 μL of protein sample to the carbon film for 60 seconds. For staining, 10 μL of 1% uranyl acetate was applied and immediately blotted off. Grids were allowed to air dry before being placed in the CM-100 TEM. Sample quality was determined by visual inspection of negatively-stained images, and quality was defined by intact particle appearance, close particle distribution and lack of obvious specimen contaminants. Samples determined to be of high quality were selected for cryo-EM analysis.

## Cryo-EM sample preparation and data collection

UltrAuFoil (Quantifoil) R1.2/1.3 300 mesh gold grids were negatively glow discharged in a Pelco EasiGlow machine for 30 seconds at 30 mA before 3 μL of purified glycoprotein sample was applied at a concentration of 0.5 mg/mL at 4˚C and 100% humidity in a FEI Mark IV Vitrobot. A wait time of 30 seconds was imposed prior to blotting the sample for 5 seconds before grids were plunge frozen into liquid ethane. Data were collected using EPU software on a FEI Titan Krios G3i (Thermo Scientific) operating at 300 kV with a Gatan BioQuantum energy filter and a K3 camera system. Exposures were collected for 3.81 seconds with an accumulated electron dose of 67 e/$\text{Å}^2$ for WIV1 and 59 e/$\text{Å}^2$ for 007 and SZ3. Exposures were fractionated into 95 frames. The energy filter slit was set at 15 eV. Exposures were collected with a range of defoci from -0.6 to -2.0 μm at a magnification of 130,000X resulting in a nominal pixel size of 0.84 Å. Data collection, refinement and validation statistics can be found in S1 Table.

## Data processing

All data processing, including map reconstruction, local resolution estimation and map sharpening, was performed in version 3 of cryoSPARC [74]. The frames of movies were aligned, motion corrected and dose-weighted using the Patch-Based Motion Correction function and global Contrast Transfer Functions were fitted using the Patch-Based CTF Correction function. Particles were picked using automated blob picking and subjected to two rounds of 2D classing. A selection of classes were used as input templates for further template-based particle picking. Particles were subjected to 2D classing until classes with clear secondary features were produced. Particle sets from such classes were used in *ab initio* reconstruction and refined using homogeneous refinement followed by several rounds of non-uniform refinement, imposing C3 symmetry. Final maps were produced for the cCoV-007 spike at a global resolution of 2.1 Å, the cCoV-SZ3 spike at a global resolution of 1.92 Å, and the bCoV-WIV1 spike at a global resolution of 1.88 Å. Local resolution was calculated from two independent half-maps using the blocres feature of cryoSPARC. A general flow diagram of our methods can be found in S7 Fig.

## Model building and refinement

Initial models were built by mutating a previously solved spike structure (PDB accession code: 7E7B) to the amino acid sequence identity of cCoV-007, cCoV-SZ3, or bCoV-WIV1 using the SWISS-MODEL function of the Expasy web server (https://swissmodel.expasy.org/). Models were manually fit into maps using UCSF ChimeraX [75] and adjusted using a high-fidelity molecular dynamics simulation calculated in ISOLDE [76]. Manual refinement was performed in WinCOOT [77]. After model idealization, N-linked glycans were modeled in and refined using the carbohydrate module of WinCoot. Real-space refinement took place in Phenix [78] and model validation was carried out using Phenix, PDBSum [78], MolPROBITY [79], and UCSF Chimera [80]. Residue interactions, such as putative hydrogen bonds, were assessed in

PyMOL [81]. Global alignments of the amino acid sequences were performed in Geneious Prime v 2023.0.1 software [82] using the Global Alignment With Free End Gaps function using a Blosum62 Cost Matric. RMSD were calculated in ChimeraX and are reported in S3–S6 Fig.

## Supporting information

**S1 Fig. Comparison of reconstruction resolutions and map quality between SARSr-CoV glycoproteins.** (*Left*) Gold-standard Fourier shell correlation plots are computed within cryoS-PARC from two independent half-maps and report a global resolution for each reconstruction. (*Right*) Amino acid residues contained within the receptor binding motif loop (L465 –T471) are presented within the electron density, set to a standard deviation threshold level of 0.25 in all panels.
(TIF)

**S2 Fig. Coulombic electrostatic potential of SARSr-CoV glycoproteins.** Molecular surface representation of side and top views of SARSs-CoV spike proteins are coloured by Coulumbic electrostatic potential calculated in UCSF ChimeraX according to Coulumbs law: $\varphi = \Sigma \, [q_i / (\varepsilon d_i)]$. Negative potential is coloured blue and positive potential is coloured red. The SARS--CoV-2 Wuhan-Hu-1 (PDB: 6ZGE), as well as SARSr-CoV spikes pCoV-GX (PDB: 7CN8), pcoV-GD (PDB: 7BBH) and bCoV-RaTG13 spike (PDB: 6ZGF) were downloaded from the Protein Data Bank.
(TIF)

**S3 Fig. Structural and sequence based comparison of S1 domains between SARSr-CoV spike glycoprotein and SARS-CoV-2 WT.** *(a)* Needleman-Wunsch protein alignment of gly-coprotein monomer cCoV SZ3 (green), cCoV 007 (blue) and bCoV WIV1 (purple), followed by alignments of the receptor binding motif (RBM), receptor binding domain (RBD) and the N-Terminal domain (NTD) of novel glycoprotein structures against SARS-CoV-2 WT (PDB: 6ZGE). *(b)* Geneious pairwise global amino acid sequence alignment of RBM from 7 spike gly-coproteins, cCoV-007, cCoV-SZ3, bCoV-WIV1, SARS-CoV-2 WT, bCoV RaTG13 (PDB: 6ZGF), pCoV-GX (PDB: 7CN4), and pCoV-GD (PDB: 7BBH).
(TIF)

**S4 Fig. Structural and sequence based comparison of N-terminal domain between SARSr-CoV spike glycoproteins.** *(a)* Needleman-Wunsch protein alignment and root mean square deviation (RMSD) calculation of N-terminal domains (NTD) from cCoV-SZ3 (green), cCoV-007 (blue), bCoV-WIV1 (purple), SARS-CoV-2 WT, bCoV-RaTG13, pCoV-GX, pCoV-GD all shown in tan. RMSD values calculated using unpruned atom pairs *(b)* Geneious pairwise global amino acidsequence alignment of NTDs from 7 spike glycoproteins, cCoV-007, cCoV-SZ3, bCoV-WIV1, SARS-CoV-2 WT, bCoV RaTG13, pCoV-GD, and pCoV-GX.
(TIF)

**S5 Fig. Structural and sequence based comparison of receptor binding domain between SARSr-CoV spike glycoproteins.** *(a)* Needleman-Wunsch protein alignment and RMSD cal-culation of receptor binding domains (RBM) from cCoV-SZ3 (green), cCoV-007 (blue), bCoV-WIV1 (purple), SARS-CoV-2 WT, bCoV-RaTG13, pCoV-GX, pCoV-GD all shown in tan. RMSD values calculated using unpruned atom pairs *(b)* Geneious pairwise global amino acid sequence alignment of receptor binding domains from 7 spike glycoproteins, cCoV-007, cCoV-SZ3, bCoV-WIV1, SARS-CoV-2 WT, bCoV RaTG13, pCoV-GD, and pCoV-GX.
(TIF)

**S6 Fig. Structural and sequence based comparison of receptor binding motif between SARSr-CoV spike glycoproteins.** *(a)* Needleman-Wunsch protein alignment and RMSD calculation of receptor binding motif (RBM) from cCoV-SZ3 (green), cCoV-007 (blue), bCoV-WIV1 (purple), SARS-CoV-2 WT, bCoV-RaTG13, pCoV-GX, pCoV-GD all shown in tan. RMSD values calculated using unpruned atom pairs *(b)* Geneious global amino acid sequence alignment of RBM from 7 spike glycoproteins, bCoV-WIV1, cCoV-SZ3, cCoV-007, SARS-CoV-2 WT, bCoV RaTG13, pCoV-GX, pCoV-GD.
(TIF)

**S7 Fig. Image processing and reconstruction of cCoV-007 glycoprotein.** (*a*) Data processing included patch-based motion correction and patch-based CTF correction. (*b*) Particles were picked by automated blob picking, extracted and classed to produce templates used in template-based particle picking. Following iterative 2D classing, 774,883 particles were selected for use in downstream reconstruction jobs. (*c*) *Ab initio* reconstruction produced an initial 3D model. (*d*) The initial model was refined in iterative homogeneous and non-uniform refinement jobs using the particle stack from (*b*) to produce a final model extending to a global atomic resolution of 1.92 Å. (*e*) An initial model was built using a reference model (PDB: 7E7B) and fit into the electron density using ISOLDE. (*d*) Refinement was performed using Stand-Alone Coot and Phenix.
(TIF)

**S1 Table. Cryo-EM data collection, refinement and validation statistics for SARSs-CoV glycoproteins.**
(TIF)

# Acknowledgments

We acknowledge the Kai Tahu peoples, the traditional guardians of the land upon which the University of Otago stands, and the indigenous Dharawal peoples, who are the traditional custodians of the land upon which the University of Wollongong stands today. We acknowledge Alexander McLellan for discussions relating to the project and laboratory space for transfection protocols. We acknowledge the use of the Otago Micro and Nano Imaging Electron Microscopy Centre at the University of Otago and the University of Wollongong Cryogenic Electron Microscopy Facility at Molecular Horizon.

# Author Contributions

**Conceptualization:** Francesca R. Hills, Yi-Ping Li, Laura N. Burga, Mihnea Bostina.

**Data curation:** Francesca R. Hills, Alice-Roza Eruera.

**Formal analysis:** Francesca R. Hills, Alice-Roza Eruera.

**Investigation:** Francesca R. Hills, Alice-Roza Eruera, James Hodgkinson-Bean.

**Methodology:** Francesca R. Hills, Alice-Roza Eruera, Fátima Jorge, Richard Easingwood, Simon H. J. Brown, Yi-Ping Li, Laura N. Burga.

**Resources:** Richard Easingwood, Simon H. J. Brown, James C. Bouwer, Mihnea Bostina.

**Software:** Francesca R. Hills, Alice-Roza Eruera, James Hodgkinson-Bean.

**Supervision:** Mihnea Bostina.

**Validation:** Francesca R. Hills, Alice-Roza Eruera.

**Visualization:** Francesca R. Hills, Alice-Roza Eruera, James Hodgkinson-Bean.

**Writing – original draft:** Francesca R. Hills, Alice-Roza Eruera.

**Writing – review & editing:** Francesca R. Hills, Alice-Roza Eruera, James Hodgkinson-Bean, Laura N. Burga, Mihnea Bostina.

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
