## [Decision Letter · Decision Letter 0]

10 Mar 2024

Dear Dr. Bostina,

Thank you very much for submitting your manuscript "Variation in structural motifs within SARS-related coronavirus spike proteins" for consideration at PLOS Pathogens. As with all papers reviewed by the journal, your manuscript was reviewed by members of the editorial board and by several independent reviewers. The reviewers appreciated the attention to an important topic. Based on the reviews, we are likely to accept this manuscript for publication, providing that you modify the manuscript according to the review recommendations.

Dear authors,

The reviewers request minor corrections. Please look through them and address line-by-line. We hope to see your resubmission soon!

Sincerely,

Shee Mei Lok, PhD

Guest Editor

PLOS Pathogens

Ashley St. John

Section Editor

PLOS Pathogens

Michael Malim

Editor-in-Chief

PLOS Pathogens

orcid.org/0000-0002-7699-2064

Dear authors,

The reviewers request minor corrections. Please look through them and address line-by-line. We hope to see your resubmission soon!

Reviewer Comments (if any, and for reference):

Reviewer's Responses to Questions

**Part I - Summary**

Reviewer #1: The manuscript from Hills et al report high-resolution cryo-EM structures of spike proteins from three SARS-related animal coronaviruses: bat SL-CoV WIV1 and civet cCoV-SZ3 and cCoV-007. The structures are at particularly high resolutions (1.88–2.10 A), among the highest ever determined. This allows the authors to observe the water networks throughout various parts of the spike proteins, including interactions with the glycans and bound fatty acid molecules. The comparative analyses are informative, providing insights into the conserved structural elements. The manuscript is well written and easy to read, and the figures clearly display the points described in the text. A few minor revisions are needed prior to publication, listed below.

Reviewer #2: Given the limited availability of structural information pertaining to closely related genetic species within SARS-related coronaviruses, this manuscript elucidates the essential structural characteristics of the Spike protein in three novel SARSr-CoVs (bat SL-CoV WIV, cCoV-SZ3, cCoV-007) by utilizing cryo-EM. The article provides a meticulous description of the intricate glycan tree on the glycoprotein, water molecule density, and fatty acid binding pocket structure. These structural features play a pivotal role in determining the functionality and stability of the Spike protein. Furthermore, the authors analyze structural disparities associated with receptor binding (ACE2), which holds paramount importance for comprehending SARS-CoV-2 origin, devising epidemic prevention strategies, and advancing therapeutic advancements.

However, there are still areas for improvement in the manuscript. Firstly, the introduction to the research background is excessively concise. It is recommended that the authors provide more detailed background information and explicitly state the research objectives. Secondly, there are numerous ambiguities and deficiencies in result interpretation, particularly regarding the inadequate validation of functional and stability hypotheses, which could significantly undermine the credibility and persuasiveness of the research. Additionally, discrepancies between the descriptions in the text and figures, along with a lack of proper explanations of figure content within the text, make it challenging for readers to comprehend the research findings. Finally, inconsistencies in capitalization and font color used in figures detract from overall formatting coherence of this paper. Standardizing these aspects will enhance readability and professionalism.

**Part II – Major Issues: Key Experiments Required for Acceptance**

Reviewer #1: (No Response)

Reviewer #2: 1. In line 40, “Further, considering that three coronaviruses emerged zoonotically in the last……”, Which the three coronaviruses are referred to here?

2. In line 62, “As both bats and civets are potential intermediate hosts for SARS-CoV-2, this research narrows the structural knowledge gap on SARSr-CoV spikes, and informs future investigations into the origin of SARS-CoV-2.” There should be corresponding references cited here.

3. In line 69-72, “Several structural motifs were proposed to regulate this behavior in the case of SARS-CoV-2 spike” refers to which structural motifs? “We have chosen three SARSr-CoVs from……” , how were the three SARSr-CoV selected? The rationale for their selection is not clearly articulated.

4. In line 80, “WIV1 share the most structural similarity with SARS-CoV-2, with RMSD values 2.3 A, 2.0 A and 2.2 A”, The values 2.3 A, 2.0 A, and 2.2 A described are not found in the figures. WIV1 and SARS-CoV-2 are not the most similar, and the RMSD value does not indicate greater structural similarity. Instead, the larger the RMSD value, the less similar the structures are.

5. In line 86,“Several non-conserved residues have altered the hydrogen bonding network that…….” Regarding the hydrogen bonding network described in the manuscript, it is not shown in the figure.

6. There are numerous inconsistencies between the Figures and the descriptions provided in the manuscript. Additionally, some Figures lack accompanying text descriptions. The order of Figures and text descriptions should be consistent.

**Part III – Minor Issues: Editorial and Data Presentation Modifications**

Reviewer #1: 1. Lines 43-44 of the introduction: this appears to be a general statement about coronaviruses, yet it states that the receptor is ACE2, which is not used by all coronaviruses. This sentence should be reworded for accuracy.

2. Table S1: list the actual number of final particle images used in the reconstruction rather than an approximation.

3. Table S1: Under “refinement”, the row “Initial model used (PDB code)” lists “ab initio”, yet in the methods the authors state “Initial models were built by mutating a previously solved spike structure (PDB accession code: 7E7B)”. Therefore, 7E7B should be listed in this row of the table.

4. A value for the map-to-model resolution should be provided, rather than listing N/A

5. There is no reason for models to have C-beta outliers. These should be fixed and re-refined.

6. PDB validation reports should be made available to reviewers to better assess the quality of the maps and models. This is now standard practice at many journals.

Reviewer #2: 1. The uppercase and lowercase letters in the figures should be consistent with those in the manuscript.

2. Some words in the manuscript are black and some are dark gray.

PLOS authors have the option to publish the peer review history of their article (what does this mean?). If published, this will include your full peer review and any attached files.

Reviewer #1: No

Reviewer #2: No

Figure Files:

Data Requirements:

Reproducibility:

References:

---

## [Editor Report · Decision Letter 1]

28 Mar 2024

Dear Dr. Bostina,

We are pleased to inform you that your manuscript 'Variation in structural motifs within SARS-related coronavirus spike proteins' has been provisionally accepted for publication in PLOS Pathogens.

Best regards,

Shee Mei Lok, PhD

Guest Editor

PLOS Pathogens

Ashley St. John

Section Editor

PLOS Pathogens

Michael Malim

Editor-in-Chief

PLOS Pathogens

orcid.org/0000-0002-7699-2064
---

## [Editor Report · Acceptance letter]

20 May 2024

Dear Dr. Bostina,

We are delighted to inform you that your manuscript, "Variation in structural motifs within SARS-related coronavirus spike proteins," has been formally accepted for publication in PLOS Pathogens.

Best regards,

Michael Malim

Editor-in-Chief

PLOS Pathogens

orcid.org/0000-0002-7699-2064